# Advances in Modelling and Analysis of Strength of Corroded Ship Structures

Krzysztof Woloszyk [1] and Yordan Garbatov [2,*]

1   Institute of Ocean Engineering and Ship Technology, Gdansk University of Technology, Narutowicza 11/12 st., 80-233 Gdansk, Poland; krzwolos@pg.edu.pl

2   Centre for Marine Technology and Ocean Engineering (CENTEC), Instituto Superior Técnico, University of Lisbon, Avenida Rovisco Pais, 1049-001 Lisboa, Portugal

*   Correspondence: yordan.garbatov@tecnico.ulisboa.pt; Tel.: +351-21-841-7907

**Abstract:** The present study reviews the recent advances in modelling and analyses the strength of corroded ship structures. Firstly, the time-variant methodologies that consider only the mean structural element thickness loss due to corrosion degradation are identified. Corrosion degradation is regarded as the phenomenon that causes uneven thinning of specimens. This has been captured by various researchers as the loss of mechanical properties of structural steel components. A review of the existing experimental and numerical studies shows significant interest in this field of study. The advances in modelling and analysis of structural behaviours of different ship structural components of larger sizes (including plates, stiffened plates and panels, and entire hull girders) are outlined. Research on the impact of general and pitting corrosion degradation is reviewed separately since the phenomena are different in terms of modelling and analysis. Additionally, recent advances concerning the reliability analysis of corroded ship structural components have also been reviewed. Finally, the general conclusions are drawn and future research topics are outlined.

**Keywords:** ship structures; strength; corrosion; degradation

## 1. Introduction

Ship structures need to be safe as well as economically justifiable. Furthermore, the possible loss of human life/cargo and environmental pollution as a result of ship sinkage should lead to even higher precautions (considering the ship's safety). Furthermore, ship sizes have increased rapidly in the last 50 years [1], leading to many structural challenges. Therefore, describing the hull's structural behaviour is an essential feature of a ship's structural design. Additionally, from the beginning of the design process, one should consider the intact state of the structure and its possible deterioration since ship structures operate in severe sea environments and are subject to severe corrosion degradation.

Corrosion could lead to significant structural capacity losses in the various components (after several years of operation). Properly estimating the deterioration level in the design stage may decrease operational expenditure since it may optimise future surveys.

At the ship design stage, to mitigate the corrosion degradation, corrosion addition and additional thickness are required by classification societies (e.g., Common Structural Rules [2]), and strength criteria are satisfied for the net thickness (neglecting the complete corrosion addition or half of it). The corrosion addition is based on the statistical analysis of thickness loss measurements, which is specific to the member location and corrosion environment taken as the 95% quantile of the predicted corrosion depth (and assuming the 25-year service life). Regular inspections are carried out throughout the ship's service life and members identified as worn are replaced. Nevertheless, the methods used to estimate structural behaviour are typically developed for intact structures and assume that the corrosion will uniformly reduce structural component thickness. When considering

the other degradation effects, acting simultaneously, such as fatigue cracking, the strength deterioration may even be magnified.

Structural problems may be crucial for ageing ships. In a recent report on maritime transport [3], the mean age of the current merchant fleet was summarised. The mean value of the age of all ships is about 21 years, including bulk carriers (9 years) and tanker ships (19 years). For general cargo ships, almost 60% of operating ships are above 20 years old, and most already exceed the typical period of exploitation. Based on the statistics of capacity loss, it is evident that the probability of loss is increasing with the ship age, as presented in the recent statistics concerning bulk carriers [4] (see Figure 1). Although most of these losses were caused due to human error (which led to, e.g., grounding) rather than pure structural failures, an ageing ship will be less resistant to such accidents.

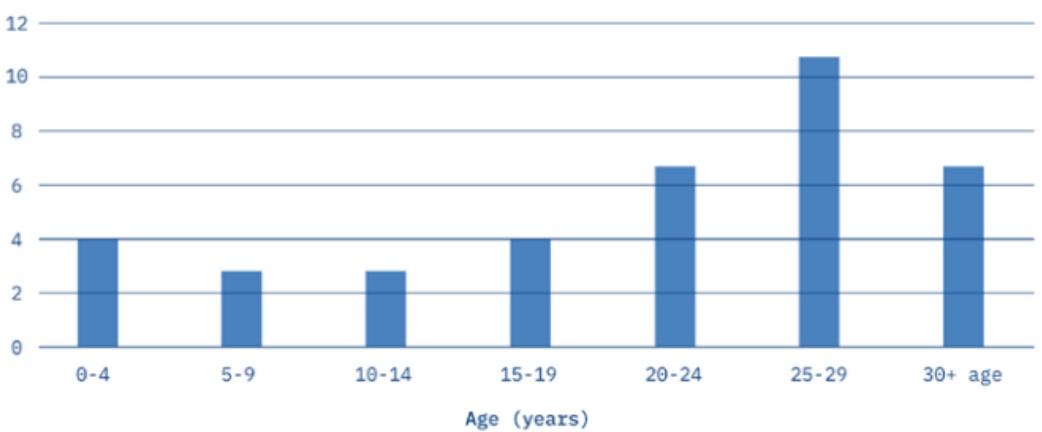

**Figure 1.** Bulk carrier's losses as a function of ship's age (2010–2019) [4].

Up to 90% of hull damages are primarily or secondary caused by excessive corrosion degradation [5]. One of the most prominent examples is the Prestige tanker ship, which broke into two parts. Regular surveys should not lead to such a situation (in theory). However, as noted in [6], there is always some probability that excessively deteriorated structural elements will be omitted during inspections. One must be aware that this probability could be of a considerable level in a bad inspection (e.g., region of double bottom).

In this view, it is essential to identify the structural behaviours of ageing structural components since their behaviours may be significantly different from the ones of intact structures. The primary ageing mechanism that needs to be considered is corrosion degradation. In the present study, we review the advances in modelling and analyse the strengths of corroded ship structures. As already outlined, the most simplified model accounts for the mean thickness loss. Thus, the existing models and analyses regarding the time-variant corrosion losses are reviewed, including recent ones.

Further, the impacts of corrosion degradation on the mechanical properties of steel specimens are discussed. Finally, both numerical and experimental studies regarding the impacts of corrosion on the strength of various ship hull members are reviewed, including both general and pitting corrosion degradation. It needs to be highlighted that these two corrosion types need to be treated differently in terms of both modelling and analysis. A summary of the recent advances regarding the modelling and analysis of corroded ship structures is given, and future research perspectives are outlined.

## 2. Time-Variant Corrosion Degradation Models

Regarding corrosion degradation modelling, the most basic concept is to account for the mean value loss of the plate thickness. This modelling could be easily adopted in many types of analyses related to the structural strengths of various components, considering both prescriptive requirements and numerical models. Many time-variant corrosion degradation models have been developed in recent years, primarily based on the statistics gathered

from specially-designed experiments or in-situ measurements in operating ships. Such models are very useful for determining the change of the mean corrosion depth with exploitation time.

Firstly, we can distinguish relatively well-established corrosion models developed mainly in the late 1990s, e.g., Melchers [7,8]; Yamamoto and Ikegami [9]; Guedes Soares and Garbatov [10,11]; Paik et al. [12]; and Qin and Cui [13]. Based on these models, developments have since been made. One of the commonly used models, developed by Guedes Soares and Garbatov [10,11], is presented in Figure 2 (left). There will be time without corrosion degradation related to the coating life at the beginning. Then, when the coating breaks, the corrosion starts to develop rapidly with the initial corrosion rate.

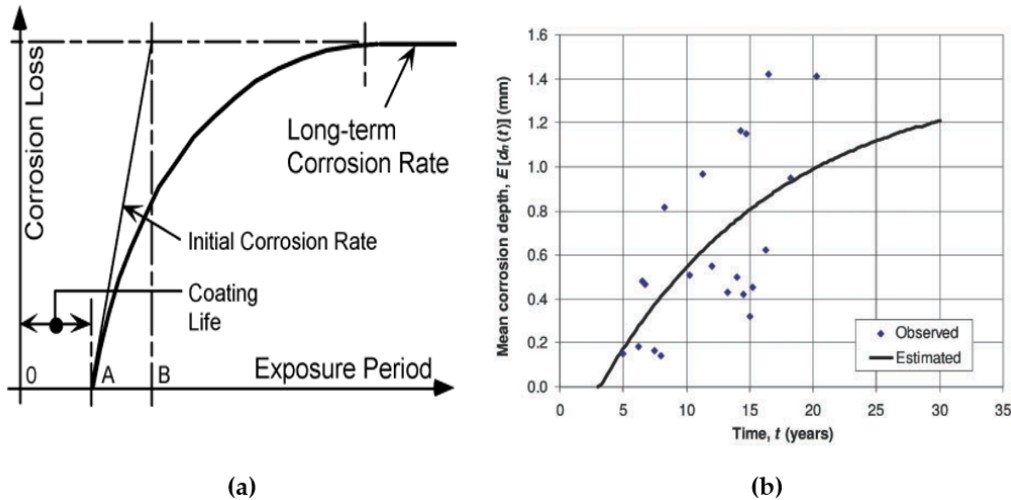

**(a)**                                      **(b)**

**Figure 2.** Corrosion degradation model of Guedes Soares and Garbatov [10] (**a**) with fitted data of the bottom plating of the bulk carriers [14] (**b**).

Further, the corrosion rate decrease and corrosion loss tend to constant values. The model parameters are typically calibrated based on the available data from measurements and can vary depending on the ship type, region, etc. Some examples of corrosion model parameters for particular ship sections could be found in the literature, e.g., deck plates of the ballast, cargo tanks of tankers [15], crude oil tanks [16], and bulk carrier decks [17]. The example of fitting the observed corrosion data—of the bottom plating of the bulk carriers into the corrosion model—is presented in Figure 2 (right).

Recently, more studies have been conduced on corrosion loss models. A simplified method to predict the corrosion levels of ship plating in different locations was presented by Zayed et al. [6]. The model accounted for different corrosion types (of each side of the considered plate) and was calibrated based on the available measurement data. Lampe and Hamman [18] presented the extended Guedes Soares and Garbatov model [11] for different structural members' corrosion predictions, which could be adjusted based on the available data. Compared to the basic model, the extended one accounts for uncertainties during the corrosion level prediction. Another model represented in the probabilistic domain was proposed by Ivosevic et al. [19] to assess the corrosion rate of fuel tank structures in bulk carriers. Based on the real measurements, it was found that the Weibull distribution well represents the corrosion rate. A similar observation was made by Mohammadrahimi and Sayebani [20], where Bayesian updating was used for time-dependent modelling of the corrosion wastage of deck panels. The corrosion model coefficients were updated based on the new data, providing a more accurate model. Kim et al. [21] also used Bayesian updating to modify the conventional corrosion wastage model of Guedes Soares and Garbatov [11]. Probabilistic corrosion depth was assessed at a given instance based on the available corrosion data. It needs to be noted that the decrease of the corrosion rate with the exploitation time is mainly related to the creation of a layer of corrosion products that prevent further degradation. Thus, cleaned plates will show an increase in the corrosion

rate. The multi-stage corrosion degradation model was proposed by Woloszyk et al. [22], where results of specimens tested in accelerated marine-immersed corrosion conditions were used for the purpose. The end-of-life corrosion statistics for medium endurance cutters were presented by Ayyub et al. [23].

The presented models could predict the mean loss of plate thickness in different ship locations, which could be helpful at the design stage. However, this will not cover all phenomena related to corrosion degradation. The obtained results could be significantly different from the real structural behaviours in terms of the strength analysis of structural elements. The approach covering other aspects will include changing mechanical properties and the non-uniform distribution of plate thickness within the single plate element.

### 3. Changes in Mechanical Properties

Recently, there have been many studies on the loss of mechanical properties of steel specimens, with the development of corrosion degradation. This includes research related to different types of corrosion, i.e., atmospheric, marine-immersed, etc. Most of the works dealt with experimental analyses, but numerical studies were also noted. The experiments covered both circular bars and flat coupon specimens. Since we will mainly find thin-walled structural members in ship structures, the studies dealing with the analyses of flat samples are reviewed herein.

One of the first studies related to that problem was performed by Garbatov et al. [24]. In this case, the specimens subjected to marine-immersed corrosion degradation, accelerated by applying electric currents, were subjected to tensile loadings. It was found that when the degradation level (considered as the percentage of mean thickness loss) exceeded the level of 20%, the loss of mechanical properties was evident. The mean residual thickness of the specimen was considered to calculate the mechanical properties. The loss of yield stress and total elongation concerning degradation level are presented in Figure 3. Notably, not only were the properties (shown in Figure 4) reduced, but Young's modulus and ultimate stress were too.

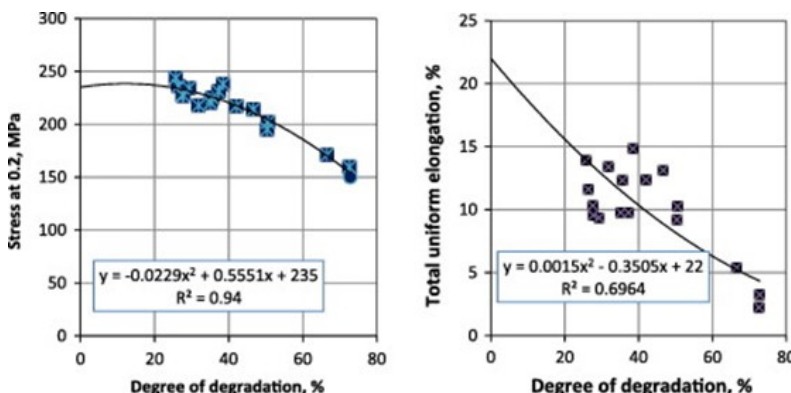

**Figure 3.** Mechanical properties as a function of degree of degradation [24].

The further experiments of that group [25,26] also accounted for the effect of specimen cleaning (e.g., sandblasting, sandpaper cleaning). The resulting mechanical properties will be higher when compared to only the corroded specimens. This leads to the hypothesis (supported by studies conducted by other researchers that performed experiments for circular bars [27–30]) that changes in mechanical properties are predominantly caused by the local non-uniformities of the corroded surfaces. In other words, if corrosion will cause perfectly uniform thinning of a specimen, significant changes in the stress–strain response of the sample should not be observed. Thus, the mechanical properties are somewhat not changed by the corrosion degradation at the material level. However, the formation of local non-uniformities causes local stress concentration points [31] that lead to the premature breaking of the specimen.

Further investigation of marine-immersed corrosion was performed by Woloszyk et al. [32], where steel specimens were subjected to accelerated corrosion degradation but without applying the electric current (the examples of the tested specimens are presented in Figure 4). In this case, the degradation level was significantly smaller than 25%. This is important since this level is typically allowed in operating structures. For more severe corrosion, structural elements should be replaced. It was found that even in lower degradation levels, there is evident loss of mechanical properties (for the degradation level of 25%, the yield stress was reduced by approximately 10%), except for Young's modulus. Additionally, the area reduction factor (related to the minimum cross-sectional area along with the specimen) was more closely associated with reducing mechanical properties compared to the degree of degradation.

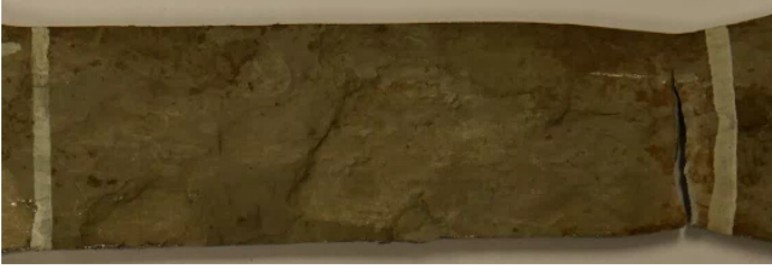

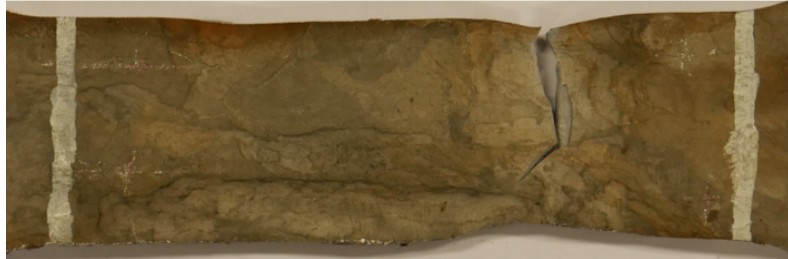

**Figure 4.** Tested specimens [32].

The ship structures are subjected to marine-immersed corrosion and atmospheric corrosion. In recent years, there have been noted works that treated that problem. Wang et al. [33] tested the specimens cut from the atmospherically corroded truss structure and observed decreased mechanical properties. However, the reduction was not so significant compared to marine immersed corrosion. An interesting study was presented by Wu et al. [34], where specimens exposed to real atmospheric corrosion, were laboratory-accelerated (in a salt-spray chamber), and subjected to tensile testing were compared. It was found that mechanical properties will always be lower than un-corroded material, but no significant difference between natural and accelerated corrosion was found.

Further, the detailed measurements using a scanning electron microscope (SEM) were performed, as presented in Figure 5 for non-cleared specimens. Two layers of corrosion products were distinguishable. The iron oxide forms the outer layer and does not transfer any loading. However, the inner layer is between the primary material and the outer layer and could transfer some loads, but its thickness is somewhat negligible compared to the total specimen thickness. Thus, as noted before, the surface morphology and related stress concentrations (also visible in Figure 5) will primarily impact the mechanical property decrease.

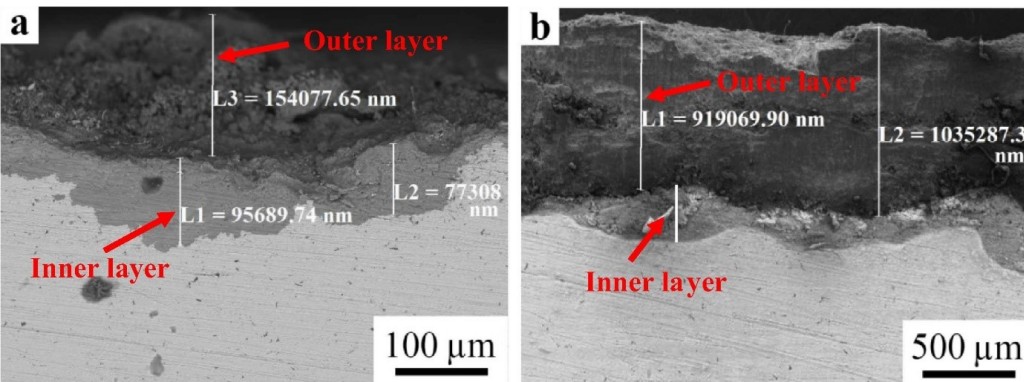

**Figure 5.** SEM images of cross-sections: (**a**) natural atmospheric corrosion; (**b**) accelerated corrosion [34].

Nevertheless, the observed finding could be significant when dealing with fatigue strength or impact strength since micro-cracks could much easier initiate in the inner layer than in the primary material. Further investigations of that problem, mainly marine immersed corrosion, are recommended. Other recent works related to the mechanical properties of specimens subjected to atmospheric corrosion can be found in [35–39]. In the case of very thin plates, such as one millimetre of thickness, the reduction of mechanical properties was down 70% from the initial values [35].

Based on experimental results, the most practical outcomes are mathematical models that predict the decrease of mechanical properties for some mean degradation level of the specimen. Such models could be easily adopted in different types of analyses, including computations performed using the finite element (FE) method. Notably, the level of reduction could vary with the corrosion type (marine immersed, atmospheric) and method of corrosion development (natural, accelerated). We can find models developed by Garbatov et al. [24], and Woloszyk et al. [32] regarding general marine immersed corrosion. The example of a model by Woloszyk et al. [32] is presented in Figure 6. There were also models regarding the impacts of pitting corrosion on mechanical properties [40,41]. However, the pitting was induced by mechanical drilling rather than from natural or accelerated corrosion degradation. Many of such models could be found for atmospheric corrosion [33,35–39,42,43]. However, until now, a unified approach that could account for changes in mechanical properties when dealing with corrosion degradation in general has not been developed.

Apart from the experiments, numerical analyses using steel specimens subjected to tensile loadings could also be performed. Typically, FE computations using explicit dynamic solvers were used. Although quasi-static conditions are typically satisfied during the experiment, the implicit solver will have problems with convergence due to multiple stress concentration points. As input for the numerical analysis, one needs to know the corroded surface's actual topography. Since more advanced measuring techniques are now available, 3D corrosion morphology detailed measurements are possible (see [22,44–46]). Wang et al. [33] implemented real scans of corroded surfaces into an FE model of tensed specimens and compared the obtained results with the experiment. Based on that, the stress concentration due to corrosion morphology was evaluated (see Figure 7). In this case, the strain distribution in the experiment was assessed via Digital Image Correlation, which allows determining the whole field of the strain distribution. The results were similar for both experimental and numerical analyses regarding strain distribution and mechanical properties. Similar numerical studies of specimens subjected to tensile loading and corrosion degradation were performed by Xiao et al. [47].

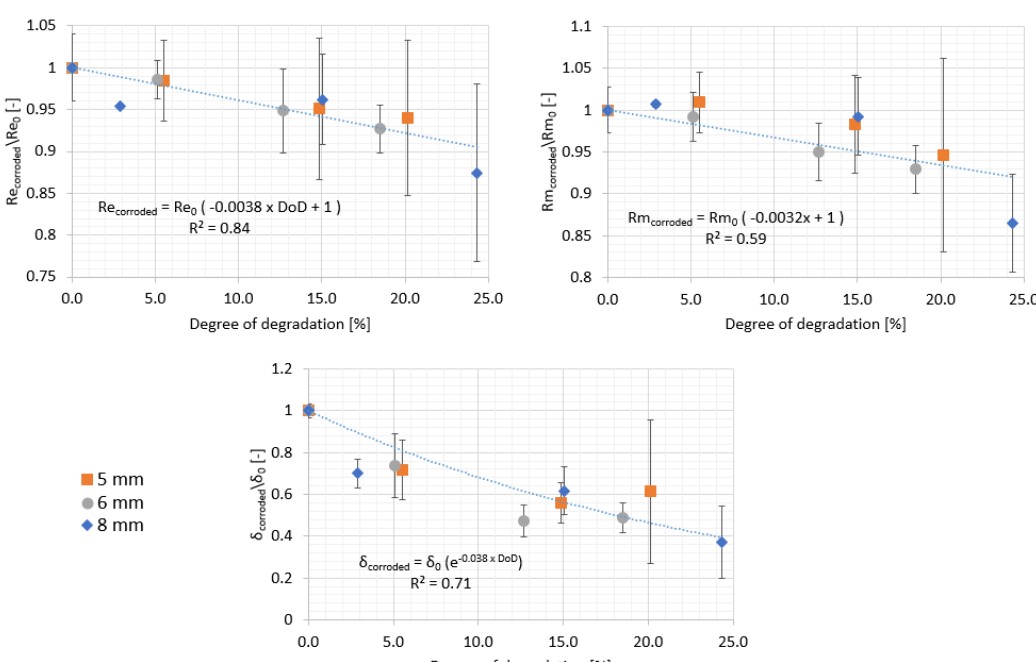

**Figure 6.** Normalised mechanical properties as a function of degree of degradation [32].

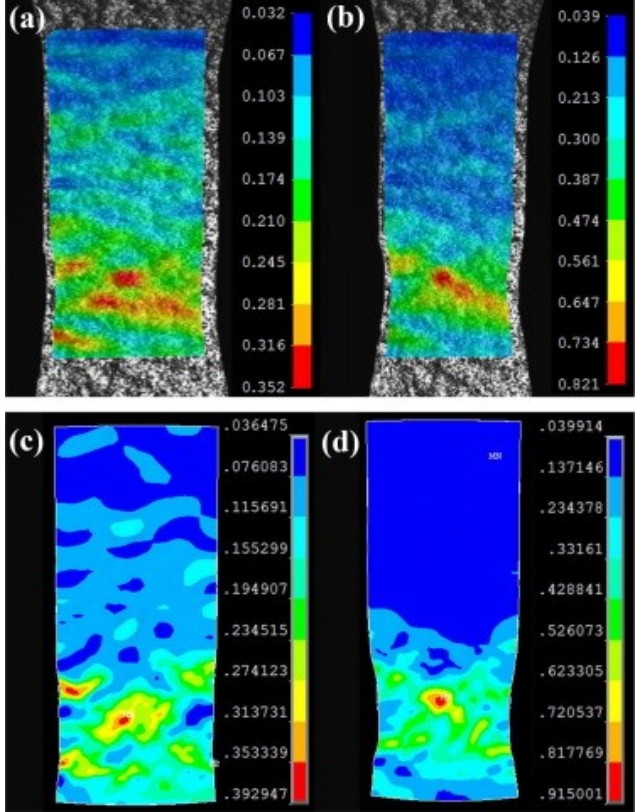

**Figure 7.** Strain distribution in a specimen subjected to tensile loading obtained experimentally (**a,b**) and numerically (**c,d**) [33].

There are disadvantages in the methodology where corrosion topography is obtained from the measurements. Firstly, it requires time-consuming corrosion testing. Secondly, usually, we can receive only a limited number of samples. However, the stochastic phenomenon of corrosion degradation requires a significant number of samples to find more general constitutive laws. The methodology that employs random field modelling recently

found some applications to model and analyse the corroded specimens. Woloszyk and Garbatov [48,49] used random field modelling to model the surfaces of the corroded specimens. The example of the FE specimen with the randomly generated field of corrosion is presented in Figure 8.

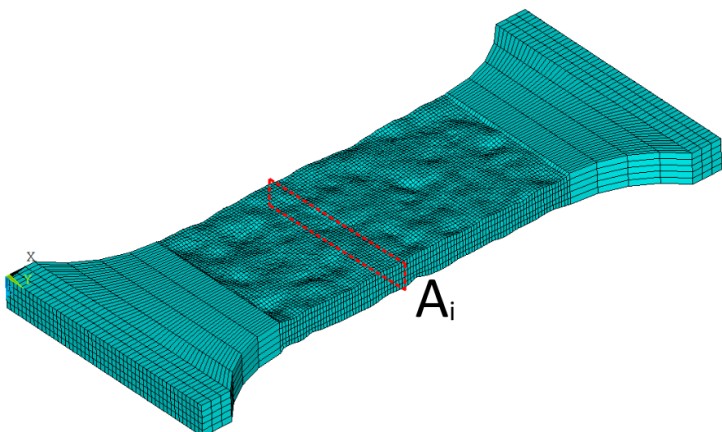

**Figure 8.** Finite element model of a specimen with a randomly generated corroded surface [49].

Various parameters of the random fields were analysed through a sensitivity analysis to find the impacts on the mechanical properties. The results of the numerical simulations were compared with the available experimental data. It was found that when statistical parameters (standard deviation, correlation level) are similar to the scanned surfaces, the mechanical properties of the corroded specimens are comparable with the experiment. The stochastic constitutive model of structural steel based on random field modelling was developed by Wang et al. [50]. In this case, numerous simulations with randomly generated fields of corrosion were carried out, leading to information about the behaviours of steel specimens corroded in the general atmosphere and offshore atmosphere corrosion conditions.

## 4. Strength Behaviour of Various Structural Components

The studies discussed in the previous section were devoted to the impacts of corrosion on the structural strength of coupon samples concerning changes in mean stress–strain response. However, this will usually not allow drawing more general conclusions regarding the structural behaviours of higher-order components. Typically, the research efforts were separated regarding the impacts of general corrosion or pitting corrosion on the strengths of various orders of structural components. Thus, the presented section reviews the advances in both outlined directions.

### 4.1. Pitting Corrosion Degradation

We should note that most works were analysed members with mechanically-induced pits rather than plates with real corrosion degradation. The primary reason could be that it is rather hard to obtain pitting corrosion in the laboratory, which requires specific conditions. Thus, the statistics obtained from pitted plates were usually used for the analysis (such statistics could be found, e.g., in [40]).

One of the first works that addressed the impacts of pitting corrosion degradation on the strengths of plates was conducted by Paik et al. [51]. The artificially-pitted plates were subjected to compressive loadings, and a significant reduction of residual strength capacity was observed. The smallest cross-sectional area was proposed as the variable that could characterise the ultimate strength value of the pitted plate. Further investigations of that group also focused on the shear strengths of the pitted plates [52]. The experimental and numerical studies regarding compressed plates taken from hold frames of bulk carriers

and pitted artificially were shown in [40]. The compressive buckling strength was smaller or equal to the plate corroded uniformly with the same degradation level.

Further, most works investigating the pitting corrosion impacts on the strengths of the plates were purely numerical. Different pit shapes were assumed in modelling, i.e., rectangular [53,54], conical [41,55], cylindrical [55–57], semi-spherical [55,58,59], or similar to real ones [40,60,61]. The pit distribution was modelled as regular or random. In terms of FE modelling, either shell or solid elements were used. Ok et al. [62] adopted artificial neural networks (ANNs) to evaluate the ultimate strengths of the compressed pitted plates. Wang [63] recently adopted a cellular automaton (CA) to simulate the plate's pit formation and analysed its ultimate strength. The methodology was fully stochastic in its origin; thus, the distributions and shapes of the pits were very close to those obtained.

Apart from most works devoted to analysing pitted plates, other structural members were also investigated (stiffened plates and panels, box girders). The experimental results, in this case, are very limited. Zhang et al. [64] performed experiments on plates with two stiffeners attached and analysed different pit dimensions and locations. There was an observed reduction in the ultimate strength and deviation of the failure mode. The numerical models of the compressed stiffened plates with pitting corrosion damage were created by Shi et al. [65] and Zhang et al. [66]. Some simplified formulations for the ultimate strength calculation were proposed. Another numerical investigation involving pitted stiffened plates and panels could be found in [53,67,68]. The impact of pitting corrosion into the torsional strength of a ship hull girder with a large deck opening was investigated via FE computations by Feng et al. [69]. Such torsional strength could be necessary in the case of, e.g., containerships. Both regular and random distributions of corrosion pits were investigated, leading to different empirical formulations. The impact of pitting corrosion on the entire hull girder's ultimate strength was investigated by Piscopo and Scamardella [70] using an incremental–iterative approach. The relation between the ultimate strength of single plating elements and the pitting degree was developed earlier in [71]. They compared hull girder ultimate strength results using an iterative method with detailed FE computations; the proposed methodology was found to be very useful, especially in the design stage.

### 4.2. General Corrosion Degradation

Unlike pitting corrosion, where mainly localised degradation occurs (although it could be widespread within an element), general corrosion [72] causes degradation on two levels. First, the mean thickness of the plating or another structural component is reduced globally. It was already outlined that the time-variant corrosion degradation models typically capture this phenomenon. In terms of diagnostics, the ultrasonic thickness gauge measurements are performed throughout the ship's service life, as required by classification societies [73]. In the second level, the corrosion causes local irregularities on the surfaces of corroded elements. Ultrasonic measurements could capture these disturbances, but the required measurement grid needs to be very dense (see Figure 9). In practice, such detailed measurements are not performed (at least considering the required measurements of classification societies). Further, there are micro-level irregularities that cannot be captured via such measurements since their sizes are smaller than the sizes of the measuring probes. This could be accounted for by changes in mechanical properties, which were broadly discussed in the previous sections.

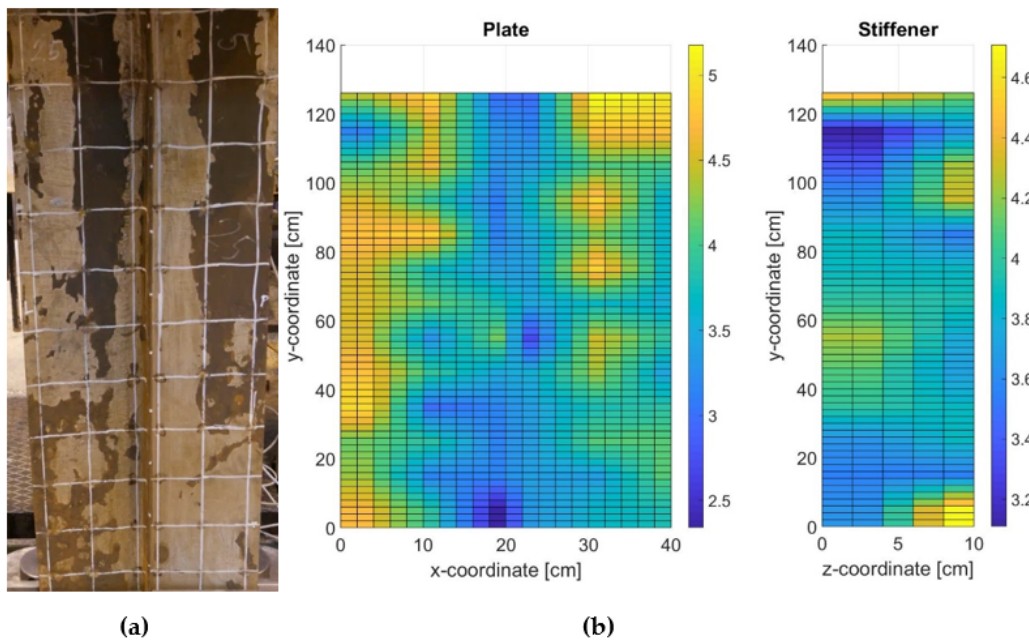

**Figure 9.** Example of a stiffened plate (**a**) within thickness distribution (**b**) in [mm] [22].

We can classify two main general corrosion degradation models, i.e., uniform and non-uniform [58]. The advantages of the uniform model were already discussed in previous sections; the main one is applicable to various types of structural analyses. On the other hand, we will have a non-uniform model with or without considering changes in mechanical properties. However, such modelling could be adopted only in FE computations, and quite a dense mesh is required. The main advantage is the more realistic structural behaviour compared to uniform modelling.

The primary object of investigation (regarding the impact of general corrosion degradation on the strength of various structural members) was 'unstiffened' plates. The numerical studies regarding the ultimate strength of corroded plates by applying a uniform model could be found in [74,75]. On the other hand, the non-uniform model of corrosion was applied by Silva et al. [58], where the distribution of plate thickness was generated with the use of the Monte Carlo simulation. Other similar works could be found [76–80], where different approaches based on random theories were adopted. In all of the analysed cases, the changes in mechanical properties were not considered. Nevertheless, significant strength reductions were observed; a higher reduction was noted when the non-uniform corrosion degradation model was applied.

There were very few experiments conducted on generally-corroded structural elements subjected to various types of loading. The experimental and numerical investigations of the ultimate strengths of stiffened plates were presented by Garbatov et al. [81], where different corrosion levels were investigated (exceeding 40% of the degree of degradation), showing significant strength reductions. The tested specimens were corroded electrochemically by immersing them in natural seawater and applying an electric current. A numerical study of the compressed stiffened plates was performed by Woloszyk et al. [82,83]. The proposed corrosion model was uniform; however, changes in mechanical properties were accounted for based on the tensile tests [24]. The numerical results were compared with experiments [81], and a good agreement was achieved. Other numerical studies of compressed stiffened plates could be found in [84–86], where both uniform and non-uniform models were adopted, but without considering changes in mechanical properties.

The extensive experimental campaign of the ultimate strength of corroded box girders subjected to the pure vertical bending moment was conducted by Saad-Eldeen et al. [87–93]; the box girders were corroded in natural seawater with the acceleration by application of

an electric current, as described in [94]. The observed reduction of capacity was significant and the structural behaviour was highly asymmetrical due to the uneven distribution of thickness reduction.

Based on the experimental results of corroded box girders and dimensional theory, Garbatov et al. [95] performed the estimation of 'hull girder ultimate strength'. It was found that current rules of classification societies overestimate the strength of the corroded ship hull since only mean thickness reduction is considered. Other studies also accounted for the effect of the general corrosion on the ultimate strength of hull girders [96–103] of various ship types (mostly tankers and bulk carriers). However, only mean thickness reduction was considered due to corrosion degradation in all cases. Considering 20 years of exploitation time, the observed loss of capacity was between 4% and 20%. Recently, Liu et al. [104] performed a numerical analysis of the crashworthiness of corroded ship hulls in stranding, accounting for uniform thickness reduction and changes in mechanical properties. The observed decrease in absorbed energy was significant since, as discussed earlier, total elongation was also reduced due to general corrosion.

## 5. Reliability of Corroded Ship Structures

Based on the previous sections, it is evident that corrosion is a stochastic phenomenon in its origin. Thus, the strengths of corroded components cannot be treated as deterministic values but rather random variables. Similarly, the acting loading is a random variable rather than a deterministic quantity due to many uncertainties accounted for in its determination. In this view, there will always be some probability that structural components will fail. From the design point of view, we can only make this probability as minimal as possible. The structural reliability analysis is typically performed considering a specific design lifetime (for ships commonly regarded as equal to 25 years). Since corrosion is a time-variant phenomenon, the reliability will decrease with exploitation time. Therefore, the structure should perform its intended function until the end of its exploitation life. In another view, the reliability analysis could decide between the possible repair or replacement of structural elements at a particular point in time. A time-variant reliability assessment is performed to see how the reliability index will change in time, which has an advantage over time-invariant solutions.

One of the first studies related to the reliability analysis of corroded ship structures was conducted by Guedes Soares and Garbatov [105]. A time-variant formulation of the reliability of maintained ship hull plates was conducted. In this case, uniform thickness reduction due to corrosion degradation was considered. It was found that considering no correlation between the corrosion rates of neighbouring elements leads to unconservative solutions. The time-variant reliability analysis of the corroding ship hull also considering the mean loss of thickness was performed by Wirsching et al. [97]. The semi-empirical formulation obtained from non-linear FE solutions was adopted to study the reliability of corroded plating by Teixeira and Guedes Soares [106]. Due to that, the non-uniformity of corrosion degradation was accounted for implicitly in the reliability formulation. Typically, a reliability problem could be formulated when one knows some analytical or semi-analytical solution. When dealing with pitting corrosion degradation, the time-variant reliability analysis of plates accounting for corrosion pit nucleation and propagation is presented in [107]. The model parameters were calibrated based on the current results of exposure tests. Woloszyk and Garbatov [108] provided a reliability-based formulation of compressed-stiffened plates considering corrosion degradation as uniform thickness loss and changes in the mechanical properties. It was observed that—where changes in the mechanical properties were not considered in the reliability analysis—the obtained reliability index was significantly overestimated.

The reliability analysis of ship hulls subjected to corrosion and maintenance was conducted by Zayed et al. [109], where ship loading uncertainties, random variables, and inspection events were considered. Woloszyk and Garbatov [110] presented a reliability analysis of the corroded tanker ship hull, where reliability formulation was developed

based on the experimentally obtained ultimate strength characteristics of stiffened plates and the dimensional theory. Moan and Ayala-Uraga [111] proposed the reliability-based model of deteriorating ship structures subjected to multiple environmental conditions. Other works related to the reliability of corroding ship hulls are also noted [96,112–114].

## 6. Conclusions

The presented study reviewed recent advances in modelling and analysed corroded ship structures. The primary modelling technique used in many analyses is time-variant corrosion modelling. The one developed by Guedes Soares and Garbatov [11] seemed to be most often used within the various existing models. Recently, research efforts have focused on extending this into another model to achieve the probabilistic representation of the time-variant corrosion degradation model, accounting for updating the model parameters based on the actual thickness measurements. Future research efforts should cover a broad spectrum of ship types and corrosion environments to calibrate the already proposed model. Additionally, as reported in the IACS Annual Review of 2020 [115], due to the actions of classification societies, including the enhanced survey programme, a reduction in thickness plating due to corrosion has been reduced in recent years. This should also be accounted for in the calibration of well-established time-variant models [116].

Another essential aspect that has been quite intensively investigated in recent years was the reduction of steel mechanical properties caused by corrosion degradation. The most dominant hypothesis is that it is caused by very localised surface non-uniformities leading to stress concentrations and premature breaking. In contrast, material properties outside the corrosion zone remained unchanged. However, more insight into the origin of this phenomenon is needed, e.g., by directly scanning the microstructures of corroded specimens. The research efforts mainly focused on developing relationships between degradation levels and observed properties. There is no general approach for such relations for various corrosion environments. Additionally, there is a significant level of uncertainty; future works should also account for that problem.

When dealing with structural components of larger sizes, structural elements, such as plates, stiffened plates and panels, box girders, and hull girders were investigated. The impacts of pitting and general corrosion were investigated somewhat separately. The analysed works were mainly numerical; there was a lack of extensive experimental studies. This is understandable since achieving corroded specimens requires significant time and effort. However, well-established experimental studies are needed to validate the proposed numerical models. Therefore, future research activity in that field is required. In terms of numerical modelling, more studies focused on analysing the impacts of pitting corrosion, which involved different types of models regarding pit shape, dimensions, and distribution. However, only a few studies accounted for the fully stochastic nature of that corrosion type; future works in that field could be conducted. When dealing with general corrosion, some basic numerical models were established. Nevertheless, future works could cover more aspects of the nature of this corrosion type and account for non-uniformities in corroded surfaces and changes in mechanical properties.

Finally, basic models used to analyse the reliability of corroded ship structures were proposed. However, future studies regarding proper uncertainty quantification, the basis for reliability formulation, are recommended.

**Author Contributions:** Conceptualization, K.W. and Y.G.; investigation, K.W.; writing—original draft preparation, K.W.; writing—review and editing, K.W. and Y.G. All authors have read and agreed to the published version of the manuscript.

**Funding:** This research received no external funding.

**Conflicts of Interest:** The authors declare no conflict of interest.

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
