# Peer review of "Advances in Modelling and Analysis of Strength of Corroded Ship Structures"

_jmse, doi:10.3390/jmse10060807_

Round 1
Reviewer 1 Report
The paper presents an extensive review of the recent advances in modelling and analysis of the strength of corroded ship structures.
The abstract and the introduction defines the contents and the aims of the paper properly, and the various theories presented throughout the text are well organised.
In the conclusions, the authors highlighted the most important aspects of the presented theories and gave suggestions about the future developments.
The paper is interesting and could be very useful when searching for literature regarding corroded ship structures analysis, so it is suitale for publication.
Author Response
The authors thank very much for the reviewer opinion and accepting the manuscript as it is.
Reviewer 2 Report
Advances in modelling and analysis of strength of corroded ship structures
1. The findings are sufficiently novel to warrant publication.
2. The conclusions are adequately supported by the data presented.
3. The article is clearly and logically written so that it can be understood by one who is not an expert in the specific field but represents only scientific review and no the scientific work. The work provides an important contribution to its field, consistent with the scope of the journal.
The paper is describing the actual problematics. Authors describing the impact of corrosion degradation on the loss of mechanical properties of steel structural components is identified throughout the review of existing experimental and numerical studies. Then, the advances in modelling and analysis of structural behaviour of different ship structural components (including plates, stiffened plates and panels, and entire hull girder) are outlined, including general and pitting corrosion degradation.
The article is only review of the literature and the methods.
Comments:
Row 182: Please explain what FE method is.

Author Response
An explanation with respect to Row 182 has been added to the text: “Such models could be easily adopted in different types of analysis, including computations performed using the FE (Finite Element) method.”
Reviewer 3 Report
The manuscript submitted by Krzysztof Woloszyk and Yordan Garbatov, reported on the “Advances in modelling and analysis of strength of corroded ship structures” In this work, the impact of corrosion degradation was summarized and many characterizations were listed from literatures in terms of different mechanical properties.
I believe this manuscript was very professional and the recent milestones were highlighted with specific figures. The literatures are disputed sufficiently and clearly. It only needs tiny modification before publication.
I suggest that the author just need to rephase your abstract. As abstract is the first paragraph that readers would like to know about your review paper. Please give a little bit more details of the background and what is included in your reviews such as different key words and show the catalogs/subtitles about how you will discuss about the topic which in your case is corrosion of ship structures.
Author Response
An enhanced new abstract is included in the manuscript in the reply of the reviewer:
"The present study reviews the recent advances in modelling and analysis of the strength of corroded ship structures. Firstly, the time-variant methodologies that consider only the mean structural element thickness loss due to corrosion degradation are identified. Furthermore, corrosion degradation is regarded as the phenomenon that causes uneven thinning of the specimens. This is captured by various researchers as the loss of mechanical properties of steel structural components. A review of existing experimental and numerical studies shows significant interest in that field of study. Then, the advances in modelling and analysis of structural behaviour of different ship structural components of larger size (including plates, stiffened plates and panels, and entire hull girder) are outlined. The research related to the impact of general and pitting corrosion degradation is reviewed separately since the phenomena are different in terms of modelling and analysis. Additionally, the recent advances in terms of reliability analysis of corroded ship structural components are also reviewed. Finally, the general conclusions are drawn, and future research topics are outlined."